

# Coral spawning patterns of *Acropora* across two Maldivian reef ecosystems

Margaux A.A. Monfared[1,*], Kate Sheridan[1,*], Simon P. Dixon[1], Matthew Gledhill[1] and Thomas Le Berre[1]

Reefscapers PVT Ltd, Male, Maldives
[*] These authors contributed equally to this work.

## ABSTRACT

Understanding patterns in coral reproductive biology at local and regional scales is crucial to elucidate our knowledge of characteristics that regulate populations and communities. The lack of published data on coral spawning patterns in the Maldives hinders our understanding of coral reproductive biology and limits our ability to assess shifts in reproductive phenology over time. Here we document baseline environmental cues, spawning patterns, exact timings and oocyte development of restored and wild *Acropora,* inhabiting shallow water reefs, across two Maldivian atolls. A total of 1,200 colonies were recorded spawning across the two sites between October 2021 and April 2023. These colonies represent 22 species of *Acropora*, with coral spawning observed over an extended period of eight months. This research details exact spawning times of multi-specific spawning, asynchronous spawning and 'split spawning' of *Acropora,* across multiple lunar phases; and highlights the need to consider restored colonies when discussing the sexual reproductive patterns of Maldivian *Acropora* in the future. Overall, corals spawned earlier in North Male Atoll compared with Baa Atoll. Earlier spawning events were significantly correlated with lower tide depths, wind speeds, daily precipitation and higher sea surface temperatures which helped explain inter-atoll, inter-annual, and intra-annual variations in spawning day. This study contributes to understanding sexual reproductive cycles of *Acropora* in the Maldives; knowledge that is vital for effective management of a critically endangered ecosystem in a changing climate.

Corresponding authors
Margaux A.A. Monfared,
margauxmonfared@yahoo.com
Kate Sheridan,
sheridankate0@gmail.com

## INTRODUCTION

Scleractinian corals are structural architects that form the foundation of coral reef ecosystems. Predominantly made from three-dimensional (3D) calcium carbonate structures (*Baird et al., 2015*; *Jamodiong et al., 2018*) they support 25% of all known marine inhabitants (*Bourne & Webster, 2013*). Reef building corals have a bipartite life history; initially as planktonic larvae that allows connectivity among reefs, and subsequently a sedentary adult stage (*Mayorga-Adame, Batchelder & Spitz, 2017*; *Davies et al., 2017*). This early planktonic period is a critical component to sustaining coral reef ecosystems, which have been on the decline over the past three decades due to numerous anthropogenic

pressures including climate change, pollution and overfishing (*Hughes et al., 2017*; *Hughes et al., 2018*).

Coral reproduction is a fundamental process that contributes to supporting coral reef functions and structure (*Baird et al., 2015*), and has evolved to improve survival by taking place during favourable conditions that maximise fertilisation, gamete density and predator satiation (*Foster, Heyward & Gilmour, 2018*). Understanding of coral reproductive timings is crucial to elucidate our knowledge of characteristics that regulate populations and communities, particularly through recruitment and ecosystem connectivity (*Kool, Moilanen & Treml, 2013*; *Done, Gilmour & Fisher, 2015*). Furthermore, sexual reproduction increases genetic diversity of offspring, improving the adaptation and resilience of the next generation (*Otto & Lenormand, 2002*), which in turn has important implications for coral reef management and conservation (*Richmond, 1997*; *Guest, 2008*).

The first documentation of multi-specific synchronous spawning took place in the 1980s on the Great Barrier Reef (*Harrison et al., 1984*). This phenomenon led to the increased effort to document coral spawning across a wide geographic region (*Harrison, 2011*). Researchers discovered that multi-specific coral spawning is probably a characteristic of all speciose coral assemblages, occurring at both high and low latitudes (*Guest et al., 2005*; *Baird, Guest & Willis, 2009*; *Bauman, Baird & Cavalcante, 2011*; *Chelliah et al., 2015*; *Bouwmeester et al., 2015*; *Gouezo et al., 2020*) and has revealed that the dominant spawning pattern of coral species is classified as hermaphroditic broadcast (*Harrison, 2011*). Local and regional environmental conditions have been shown to regulate the seasonal timing of gametogenic cycles either as ultimate factors or proximate cues (*Babcock et al., 1986*; *Harrison & Wallace, 1990*) and typically include sea surface temperature (*Keith et al., 2016*; *Sakai et al., 2020*), wind speed (*Van Woesik, 2010*; *Keith et al., 2016*; *Sakai et al., 2020*; *Lin & Nozawa, 2023*), precipitation (*Hayashibara et al., 1993*; *Mendes & Woodley, 2002*) and lunar phase (*Brady, Hilton & Vize, 2009*; *Boch et al., 2011*; *Kaniewska et al., 2015*; *Lin et al., 2021*), among others.

Located in the Indian Ocean lies the Maldives (3.2028°N, 73.2207°E), in which coral reef ecosystems were subjected to two mass bleaching events in 1998 and 2016 (*Cowburn et al., 2019*). Coral coverage decreased from 40–60% to less than 8% after the 1998 bleaching event (*Morri et al., 2015*; *Pisapia et al., 2016*). Despite this disturbance, Maldivian reefs showed resilience and recovered up to 40% in 2015 (*Pisapia et al., 2016*). However, in 2016, another mass bleaching event extirpated many reef building corals, including those of the genus *Acropora* (*Pisapia, Burn & Pratchett, 2019*). Coral bleaching has been shown to decrease reproductive potential of survivors (*Leinbach et al., 2021*), reduce gamete numbers (*Ward, Harrison & Hoegh-Guldberg, 2000*), and lead to a long-term impact on reproduction over multiple spawning periods (*Levitan et al., 2014*). To alleviate these impacts on coral reef ecosystems, restoration initiatives that manipulate asexual propagation were implemented not only in the Maldives but worldwide (*Boström-Einarsson et al., 2020*; *Montano et al., 2022*). As the persistence of coral populations rely on the success of natural recruitment through reproduction (*Richmond, Tisthammer & Spies, 2018*), it is imperative we understand reproductive patterns to predict population recovery following disturbance in the Maldives.

Although coral spawning is a well-known phenomenon where patterns in the Indo-Pacific have been collated from both the literature and unpublished observations (*Baird et al., 2021*), little remains documented about the extent of spawning synchronicity in the Maldives, the world's seventh largest coral reef ecosystem, comprising 3.14% of global reefs (*Dhunya, Huang & Aslam, 2017*). Published reports from the Maldives include observations on coral slicks in South Ari Atoll (*Loch et al., 2002*), and spawning of *Pocillopora verrucosa* (*Sier & Olive, 1994*). Mentions of *A. hyacinthus* and *A. digitifera* spawning during March and April were stated in personal communications (*Clark & Edwards, 1999*) and a published report by *Harrison & Hakeem (2007)* revealed patterns of asynchronous spawning over multiple lunar cycles. Informal records of slick formations and the inference of spawning are mentioned during March–April and in the last quarter of the year (*Marine Research Institute, 2023*). Blog posts reveal coral spawning observations in April–May 2012 at Gili Lankanfushi, North Male Atoll, (*Gili Lankanfushi Maldives, 2012*) and also in December 2014 (*Gili Lankanfushi Maldives, 2014*), whilst newsletters from Laamu Atoll have publicised spawning information over multiple months since 2021 (*Maldives Underwater Initiative, 2021*). Yet exact timings and species information is not readily available. The lack of published data to distinguish the onset of gametogenesis "*in-situ*" and exact spawning times to amalgamate regional spawning patterns hinders our understanding of coral reproductive biology at a regional scale and limits our ability to assess shifts in reproductive phenology over time. The Maldives territory is 99% water spanning across 26 atolls and is home to 258 species of hermatypic corals (*Dhunya, Huang & Aslam, 2017*). While informal spawning research has been documented through blogs and newsletters, peer-reviewed literature remains scarce. Moreover, the impact of coral restoration activities on the natural spawning cycle of *Acropora* has not been documented in this region.

Asexual propagation techniques can be limited by species availability and constrained genotypic diversity of clonal fragments (*Oppen et al., 2017*). However, the successful out-planting of vast numbers of genets of the same species can lead to 'spawning hubs' that reproduce sexually resulting in a mass supply of coral larvae released back into the environment (*Horoszowski-Fridman, Izhaki & Rinkevich, 2011*; *Montoya Maya et al., 2016*). Moreover, if donor colonies are the remnants of preceding mass bleaching events, coral propagation improves the likelihood of bleaching-resistant genotypes within populations which can be passed onto new recruits improving ecosystem resilience (*Montoya Maya et al., 2016*). Reefscapers Pvt Ltd (hereafter referred to as: Reefscapers) utilises asexual propagation through their coral frame technique by attaching coral fragments, of varying species and genera, onto metal frames which have been previously coated with resin and sand, using cable ties (*Morand, Dixon & Le Berre, 2022*). Each frame is given a unique reference code. Due to the high mortality of branching species following the 2016 coral bleaching event (*Pisapia, Burn & Pratchett, 2019*; *Bessell-Browne et al., 2021*), propagation efforts were predominantly focused on increasing coverage of the coral genus *Acropora*.

In this study we document baseline environmental cues, spawning patterns, exact spawning timings, and oocyte development of naturally occurring and restored *Acropora*

inhabiting shallow water reefs across two islands, over an 18-month period (October 2021–April 2023). This information will be critical to begin to understand coral spawning synchronicity of Maldivian Reefs at a local and regional scale and will help to direct conservation and management strategies in a changing environment.

## MATERIALS & METHODS

### Study sites

Research surveys took place around two resort islands with long-term restorative projects run by Reefscapers situated in two different Maldivian Atolls: Landaa Giraavaru located in Baa Atoll (5.2862°N, 73.1121°E) a UNESCO Biosphere Reserve since 2011 and Furana Fushi (4.2500°N, 73.5458°E), located in North Male Atoll, Maldives, since 2020 (Fig. 1). Landaa Giraavaru is 0.18 km$^2$ in area in a small sand cay situated on the western front of the Maldivian atoll chains (*Hein et al., 2020*). Furana Fushi is located 8.5 km north of Male, the capital city of the Maldives.

Coral colonies were surveyed up to an 11m depth around the 'house reef', with particular focus on the southern near-shore reef of Landaa Giraavaru. At Furana Fushi, surveys were conducted up to a 6m depth at two shallow, near-shore reef sites and two lagoon sites. The most severely affected genus from the 2016 coral bleaching event, *Acropora* (*Pisapia, Burn & Pratchett, 2019*; *Bessell-Browne et al., 2021*), was the main focus of this study. Survey sites were chosen based on *Acropora* coverage and diversity of reef type, which were: (i) Wild–naturally occurring colonies, (ii) Relocated–colonies moved from another site and transplanted directly onto the reef, (iii) Frame–colonies asexually propagated as small fragments onto Reefscapers human-made structures, and (iv) Pyramid–relocated colonies transplanted directly onto Reefscapers human-made structures in their entirety (*i.e.,* not asexually propagated). Pyramid and relocated colonies are only located at Furana Fushi and were relocated from Gulhifalhu, a reclaimed island situated in the south of the North Male Atoll and roughly 12.5km from Furana Fushi. The pyramids have been ''*in-situ*'' since June 2020.

### Surveys
#### Identification of gravid colonies
To predict the month of spawning, reproductive maturity surveys were conducted in-water for *Acropora* from September 2021–April 2023, by a minimum of two observers in each location. Surveys coincided with coral restoration activities to reduce negative impacts on coral colonies. Observers used scissors to fragment a 1–2 inch section of a coral colony from the base to avoid the infertile zone (*Randall, Giuliano & Page, 2021*), and identify the presence or absence of gametes (*Harrison et al., 1984*). Reproductive stages were tracked and classified into three distinct categories based on oocyte colouration: (i) white–immature (Fig. 2A), (ii) pale–close to maturity (Fig. 2B), and (iii) pigmented–mature (Fig. 2C), following *Baird, Marshall & Wolstenholme (2000)*. These classifications were used to inform field monitoring for spawning and were not included in statistical analysis. Upon observation of gametes, the species, location, reef type and oocyte category of gravid colonies were recorded in a central database. Gravid colonies were thereafter sampled

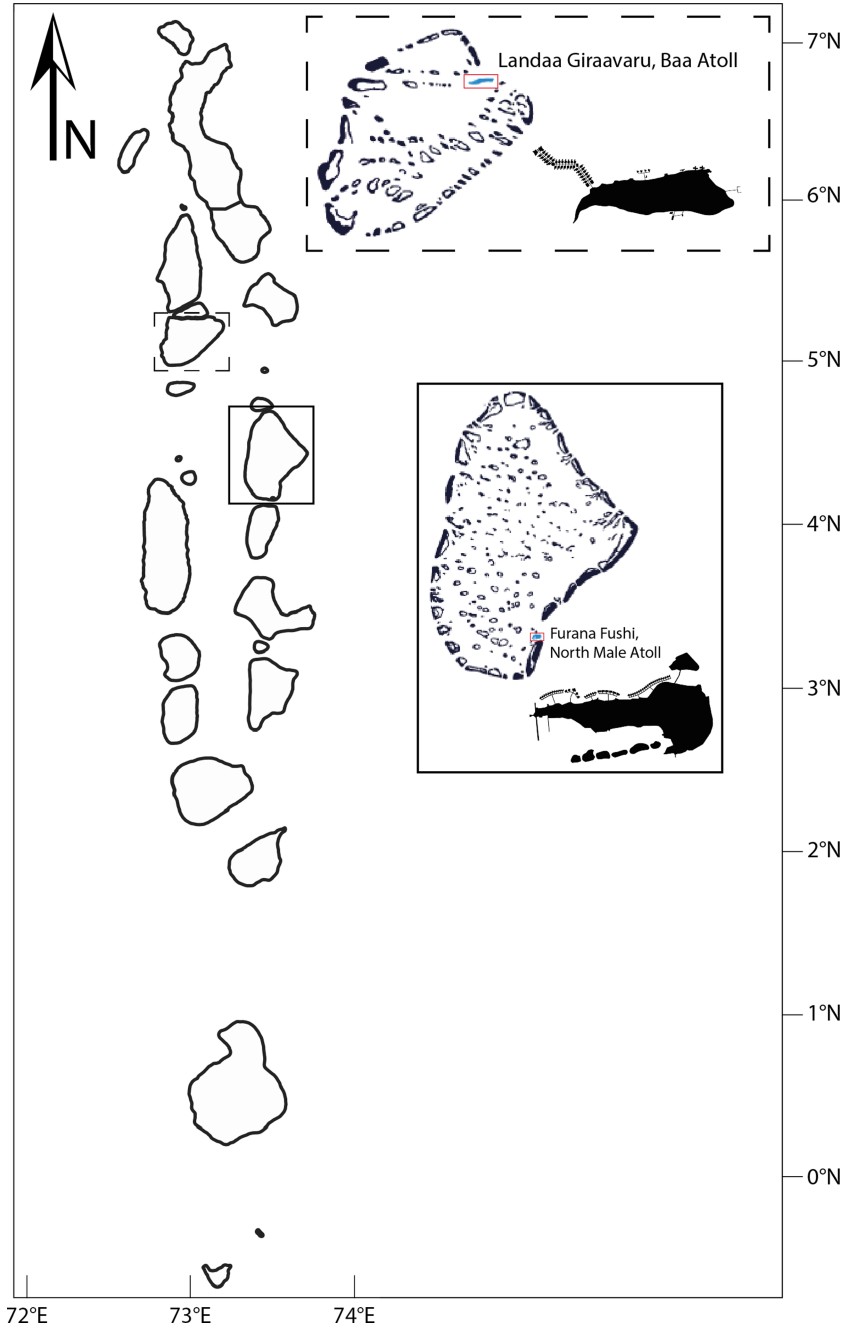

**Figure 1** Location of study sites in the Maldives: Landaa Giraavaru, Baa Atoll, and Furana Fushi, North Male Atoll.

bi-monthly to track changes in colouration. If sampling bi-monthly was not possible (due to adverse weather, staffing constraints, access to location *etc.*), colonies were sampled monthly. Coral species were identified using Corals of the World (*Veron & Stafford-Smith, 2000*). Genetic testing would need to be carried out to confirm species identification.

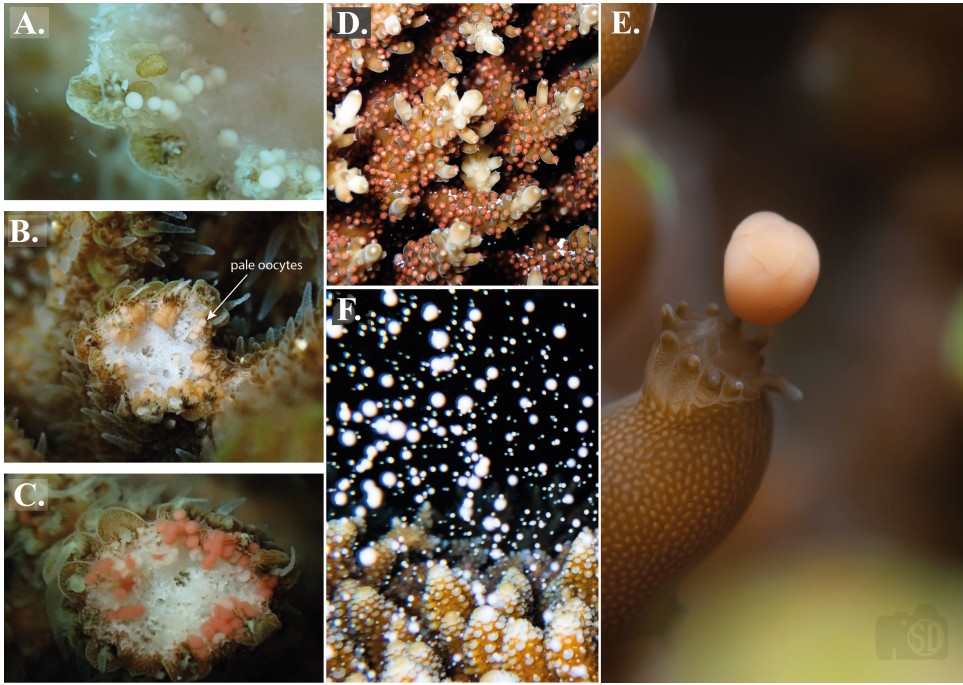

**Figure 2** **Stages of sexual reproduction in *Acropora*, Maldives.** (A–C) Development of oocyte colouration from white (immature; (A)), to pale (B), to pigmented (mature; (C)) in *A. millepora*. These classifications were used to inform field monitoring for spawning. Photographs by Margaux A.A. Monfared. (D) Observed gamete 'setting' in *A. secale* prior to spawning. The presence of 'setting' as shown in this image is what observers were looking for during nightly surveys. Photograph by Kate Sheridan. (E) Photograph to show moment of gamete bundle release in *A. rosaria*, defined as the spawning time per colony. Photograph by Simon P. Dixon. (F) A colony of *A. gemmifera* spawning egg-sperm bundles. In situ observations suggested these oocytes were white-pale. Photograph by Kate Sheridan.

### Observing coral spawning and bundles

Upon observation of gravid colonies with mature gametes within the skeletal tissue, nightly surveys took place over a period of six hours (16:30–22:30) around the full moon and new moon. Observers checked for 'setting' (tightly packed egg and sperm bundles that 'set' in the coral polyp mouth prior to being released; Fig. 2D), in previously identified gravid colonies. These bundles are buoyant due to lipid components and thus float to the surface upon release (*Arai et al., 1993*). The times of bundle appearance/setting and spawning time vary among *Acroporid* species (*Babcock et al., 1986*; *Fukami et al., 2003*); thus observers entered the water prior to sunset and remained checking for setting for up to two hours after sunset. Upon identifying setting in any colonies, observers remained in the water until bundle release to record the time of spawning. Setting time and spawning time (defined as the first observed gamete release; Fig. 2E) were recorded for individual colonies.

Surveys around full moon began two days prior and continued for up to eight days after, whilst new moon surveys took place two days prior for up to four days after, from October 2021–April 2023 at all sites. Surveyors utilised a combination of conventional SCUBA and free diving at both locations, strategically placing teams of free divers at shallow survey sites

(where depth < 3 m), and the SCUBA divers at deeper depths. Due to the placement of restored colonies adjacent to their wild counterparts we were able to determine individual spawning times to the minute. Individual surveyors remained within a specified area to ensure accuracy.

During nightly surveys, all timings were recorded on slates "*in-situ*" by observers. In the case of logistical constraints preventing night surveys, such as adverse weather, staffing numbers or site access, colonies were sampled daily and checked for absence of gametes, through reproductive maturity surveys, the following morning to ensure exact spawning dates.

The sunset time was recorded for both study sites using 'Time and Date' webpage (*Time and Date AS, 1995a*; *Time and Date AS, 1995b*), to later calculate the individual colony spawning and setting time after sunset, as sunset can be associated with spawning behaviour (*Babcock et al., 1986*; *Brady, Hilton & Vize, 2009*; *Sweeney et al., 2011*; *Keith et al., 2016*) and a useful indicator for predicting spawning (*Baird et al., 2022*). Data and information on colonies that were not previously recorded as gravid but were observed to spawn were also collected. In addition, the depth of the low tide closest to sunset (m) was collected (*Tideschart, 2023a*; *Tideschart, 2023b*) as a proxy for tide depth at the time of spawning to ensure a constant measure.

## Statistical analysis

### Variations in Spawning time and date

Differences in spawning day and time between atolls and annually for all observed colonies were tested, and for individual species where the number of colonies was greater than 30. Response variables were tested for normality using the Shapiro–Wilk test, followed by Levene's test for Equality of Variances. All response variables demonstrated a non-normal distribution, and thus Kruskal–Wallis tests were used. In instances where spawning was recorded in more than two years, Dunn's tests with Bonferroni corrections were performed to establish differences in spawning day and time for all combinations of year.

Kruskal–Wallis tests were also used with the same response variables to compare spawning amongst reef types, for *A. humilis, A. digitifera,* and *A. millepora* in Baa Atoll. Only two reef types were recorded for these species: wild and frame colonies. These species and atoll were chosen for this analysis due to sample size limitations; in North Male Atoll there were too few wild colonies observed spawning for statistical analyses, and in Baa Atoll the number of wild colonies to spawn was too small for all other species.

### Influences of environmental factors on spawning day

To assess whether potential inter-atoll or inter-annual variations in spawning day are associated with environmental conditions, Kendall's rank correlations were tested between the day of spawning relative to the nearest full moon with sea surface temperature (*Keith et al., 2016*; *Sakai et al., 2020*), wind speed (*Van Woesik, 2010*; *Keith et al., 2016*; *Sakai et al., 2020*), precipitation (*Hayashibara et al., 1993*; *Mendes & Woodley, 2002*) and tide depth (*Jamodiong et al., 2018*). Average daily sea surface temperatures (SSTs) were obtained from seatemperatures.net (*Sea Temperature, 2013a*; *Sea Temperature, 2023b*). Total daily

precipitation (mm) and average daily Wind speed (mph) were obtained from Windy.app (*Windy.app, 2023*).

The same correlations were also assessed for each atoll separately. These non-parametric approaches were chosen due to the non-normal distribution of the data and the violation of assumptions required for parametric tests. Statistical significance was determined at the $\alpha = 0.05$ level, and *p*-values less than 0.05 were considered statistically significant. All statistical analysis were conducted using RStudio version 2022.12.0 (*R Core Team, 2022*; *RStudio Team, 2022*).

## RESULTS

In total, 1,200 colonies were recorded spawning across the two sites between 1 October 2021 and 30 April 2023: 501 frame, 593 pyramid, 2 relocated and 104 wild (Table 1). These colonies represented 22 species of *Acropora*: 17 species were recorded in Baa Atoll and 19 species in North Male Atoll (Table 1). Fourteen species were recorded spawning in both atolls. Coral spawning was recorded in eight months of the year: January ($N = 1$), February ($N = 2$), March ($N = 66$), April ($N = 553$), May ($N = 2$), October ($N = 53$), November ($N = 444$), and December ($N = 79$). Furthermore, spawning was recorded on 46 days throughout the study period; 24 days of which recorded spawning of more than one species. The highest number of species recorded spawning on the same day was 14, which occurred on 5 April 2023 in North Male Atoll. Coral spawning was recorded across multiple lunar phases (Fig. 3A) and after sunset (Fig. 3B). Thirty-six colonies were recorded spawning around the new moon. Fourteen colonies spawned over more than one consecutive day within one spawning season: six *A. tenuis*, two *A. rosaria*, one *A. millepora* and five *A. nasuta*. One colony of *A. gemmifera* and two colonies of *A. tenuis* in North Male Atoll were witnessed spawning white-pale gamete bundles (see Fig. 2F).

### Oocyte development
Of the 1,200 colonies documented spawning in this study, 503 were first identified with immature oocytes across 16 species. On average, immature oocytes were identified 94.5 days before spawning.

### Inter-annual variation
There was a statistically significant annual difference in spawning day relative to full moon (K–W test; $X^2(1) = 131.51$, $\eta^2 = 0.108$, $p < 0.001$), which was consistent between all combinations of year based on a Dunn's test with Bonferroni corrections ($p < 0.001$ for all combinations). However, spawning time after sunset was not different between years (K–W test; $X^2(1) = 5.0308$, $\eta^2 = 0.003$, $p = 0.081$). When looking at species-specific spawning, of the species tested (where $N > 30$) annual variation in spawning time was statistically significant in five species, although the effect of year on spawning time for all species is small, suggesting limited ecological significance (Table 2). Of the species tested, day of spawning was significantly different annually for all species except for *A. plantaginea*. The effect of year on spawning day varied between species, and was greatest for *A. cytherea* and *A. millepora* (Table 2).

**Table 1  The number of colonies to spawn and months of spawning per *Acropora spp.* in each atoll and reef type.**

| Species | Reef type (No. Colonies) | | Months of spawning | |
|---|---|---|---|---|
| | Baa | North male | Baa | North male |
| *A. aspera* | F = 4 | P = 2 | October, November | April |
| *A. austera* | NA | P = 5 | NA | April |
| *A. clathrata* | NA | P = 1 | NA | April |
| *A. cytherea* | NA | P = 85 | NA | March, April |
| *A. digitifera* | F = 37<br>W = 19 | F = 3 | March, April | April |
| *A. gemmifera* | F = 2<br>W = 2 | F = 1<br>P = 12 | March, April | January, March, April, November |
| *A. globiceps* | W = 2 | NA | October, November | NA |
| *A. hempricii* | NA | F = 1<br>P = 15 | NA | March, April |
| *A. humilis* | F = 64<br>W = 18 | F = 5<br>P = 43 | March, April, October, November | March, April, May, November |
| *A. hyacinthus* | W = 1 | F = 2<br>P = 15 | April | March, April |
| *A. latistella* | NA | F = 1<br>P = 1 | NA | April |
| *A. millepora* | F = 29<br>W = 27 | F = 1<br>P = 19 | March, April | March, April |
| *A. muricata* | F = 3 | F = 11<br>P = 39 | April | April |
| *A. nasuta* | F = 4<br>W = 5 | P = 14 | March, April | March, April |
| *A. plantaginea* | F = 133<br>W = 8 | F = 31<br>P = 3 | October, November, December | November, December |
| *A. retusa* | F = 2 | NA | November | NA |
| *A. rosaria* | F = 6<br>W = 1 | NA | November | NA |
| *A. samoensis* | F = 6 | F = 2<br>P = 5 | April | March, April, November, December |
| *A. secale* | F = 42<br>W = 6 | F = 28<br>P = 120<br>R = 2<br>W = 8 | March, April, October, November | October, November, December |
| *A. squarrosa* | F = 6 | P = 2 | November | November |
| *A. tenuis* | F = 56<br>W = 4 | F = 18<br>P = 210<br>W = 3 | November, December | March, April, November, December |
| *A. valida* | F = 3 | P = 1 | February, May | April |

**Notes.**
Reef types are recorded as F, Frame; W, Wild; R, Relocated; and P, Pyramid.

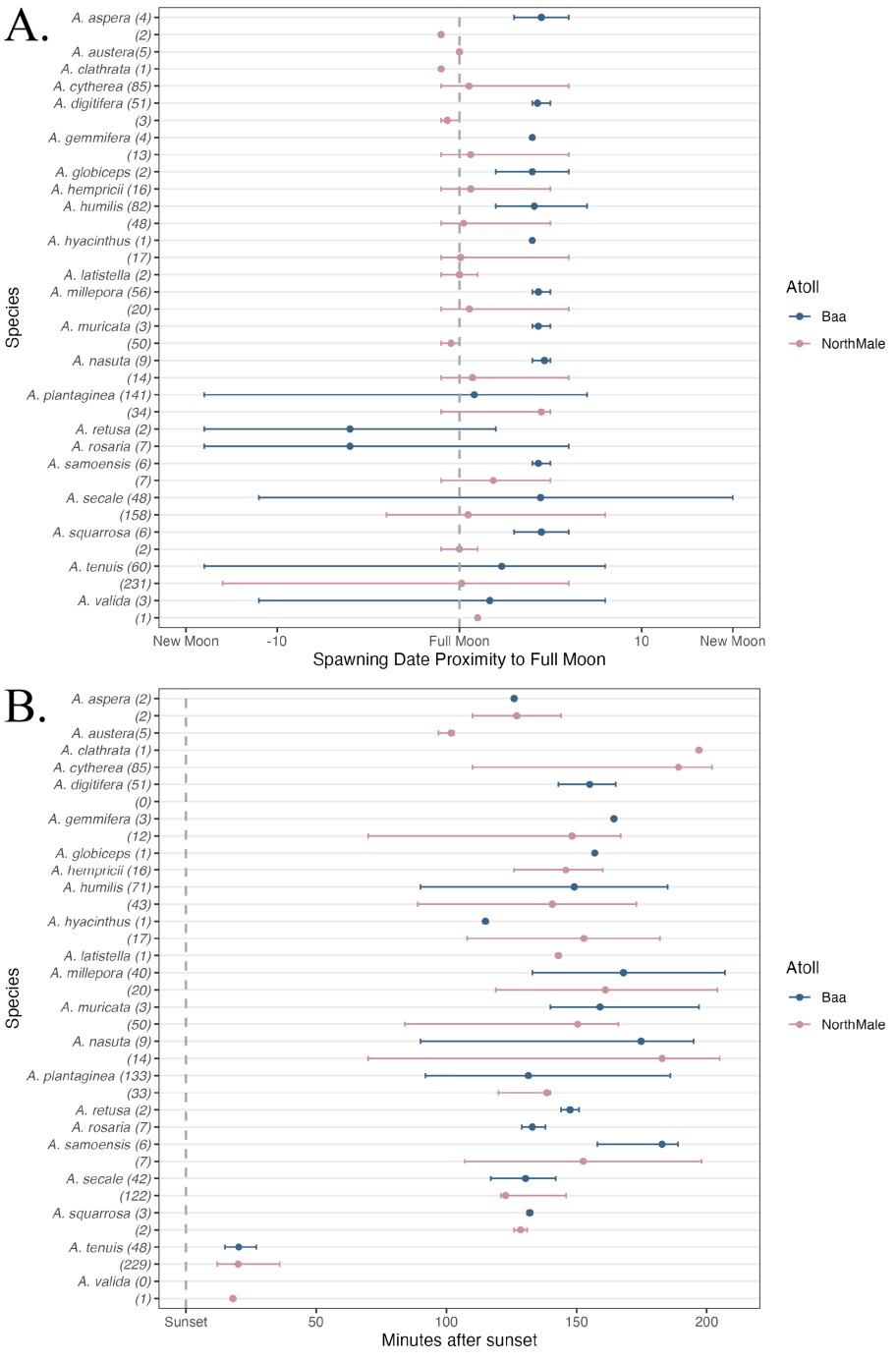

**Figure 3  Spawning date and time of *Acropora* spp.** (A) The day of coral spawning of *Acropora* spp. relative to the nearest full moon. The filled circles are the mean observed spawning day, and the bars are the range, per species. Observations from Baa Atoll are given in blue, and North Male Atoll in pink. Within the brackets after each species, the total number of colonies to spawn of that species is given. (B) The time of coral spawning of *Acropora* spp. after sunset. The filled circles are the mean observed spawning time after sunset, and the bars are the range, per species. Observations from Baa Atoll are given in blue, and North Male Atoll in pink. Within the brackets after each species, the number of colonies within that species of which an exact spawning time was recorded.

**Table 2** Results of Kruskal–Wallis tests to assess the difference in spawning day relative to full moon and spawning time after sunset between years, for *Acropora spp.*

| Species | Response | | Explanatory | | Response | | Explanatory | |
|---|---|---|---|---|---|---|---|---|
| | Spawn date relative to full moon (day) | | Year | | Spawning time after sunset (minutes) | | Year | |
| | $X^2$ | df | $\eta^2$ | $p$ | $X^2$ | df | $\eta^2$ | $p$ |
| *A. cytherea* | 52.903 | 1 | 0.625 | **<0.001** | 14.433 | 1 | 0.162 | **<0.001** |
| *A. humilis* | 22.404 | 2 | 0.161 | **<0.001** | 11.334 | 2 | 0.074 | **0.003** |
| 2021–2022 | | | | **0.047** | | | | 0.161 |
| 2021–2023 | | | | **<0.001** | | | | **0.015** |
| 2022–2023 | | | | **<0.001** | | | | **0.047** |
| *A. millepora* | 45.948 | 1 | 0.607 | **<0.001** | 0.321 | 1 | – | 0.571 |
| *A. plantaginea* | 0.330 | 1 | – | 0.5657 | 4.446 | 1 | 0.020 | **0.035** |
| *A. secale* | 18.338 | 2 | 0.081 | **<0.001** | 15.289 | 2 | 0.066 | **<0.001** |
| 2021–2022 | | | | **<0.001** | | | | **0.002** |
| 2021–2023 | | | | 1 | | | | **0.024** |
| 2022–2023 | | | | 0.725 | | | | 0.193 |
| *A. tenuis* | 61.774 | 2 | 0.208 | **<0.001** | 22.324 | 2 | 0.074 | **<0.001** |
| 2021–2022 | | | | **<0.001** | | | | - |
| 2021–2023 | | | | **<0.001** | | | | - |
| 2022–2023 | | | | **<0.001** | | | | **<0.001** |

**Notes.**
Significant results are given in bold. Values less than 0.001 (highly significant) are denoted as <0.001.

## Inter-atoll variation

There was a statistically significant difference in spawning day relative to the full moon between atolls (K–W test; $X^2(1) = 318.77$, $\eta^2 = 0.265$, $p < 0.001$). While significant differences were also found in all species tested (Table 3), the effect sizes were small-medium for all species except *A. humilis* (*Cohen, 1992*). Furthermore, spawning time after sunset was also significantly different between atolls (K–W test; $X^2(1) = 36.924$, $\eta^2 = 0.030$, $p < 0.001$). However, this was largely due to variations in coral community composition, and at the species level within species spawning time was only statistically different between atolls for *A. plantaginea* ($p < 0.001$), *A. humilis* ($p = 0.046$), and *A. secale* ($p < 0.001$), with varying effect sizes (Table 3).

## Environmental predictors of spawning

Earlier spawning events relative to full moon were significantly correlated with lower tide depths, wind speeds, and precipitation levels, but higher sea surface temperatures (Table 4). When considering each atoll individually, this trend was also true (Table 4). Tide depth had the strongest correlation with spawning day in both atolls, with the strongest correlation in Baa Atoll (Table 4). In North Male Atoll, there was little difference in the correlation coefficients of each variable, suggesting they have a similar effect on spawning day.

**Table 3** Results of Kruskal–Wallis tests to assess the difference in spawning day relative to full moon and spawning time after sunset between atolls, for *Acropora spp.* Significant results are given in bold. Values less than 0.001 (highly significant) are denoted as <0.001.

| Species | Response | | | | Response | | | |
|---|---|---|---|---|---|---|---|---|
| | Spawn date relative to full moon (day) | | Explanatory Atoll | | Spawning time after sunset (minutes) | | Explanatory Atoll | |
| | $X^2$ | df | $\eta^2$ | $p$ | $X^2$ | df | $\eta^2$ | $p$ |
| A. digitifera | 12.564 | 1 | 0.203 | **<0.001** | *Timing not recorded.** | | | |
| A. humilis | 71.094 | 1 | 0.548 | **<0.001** | 3.977 | 1 | 0.023 | **0.046** |
| A. millepora | 20.383 | 1 | 0.262 | **<0.001** | 0.001 | 1 | – | 0.975 |
| A. muricata | 10.598 | 1 | 0.188 | **0.001** | 0.014 | 1 | – | 0.907 |
| A. plantaginea | 21.198 | 1 | 0.117 | **<0.001** | 40.358 | 1 | 0.228 | **<0.001** |
| A. secale | 56.381 | 1 | 0.271 | **<0.001** | 82.551 | 1 | 0.400 | **<0.001** |
| A. tenuis | 70.050 | 1 | 0.239 | **<0.001** | 0.039 | 1 | – | 0.843 |

Notes.

*In North Male Atoll, *A. digitifera* was recorded spawning through absence of gamete surveys. Therefore, the exact spawning date was obtained but not the spawning time.

**Table 4** Kendall's rank correlations and corresponding effect sizes assessing the influence of local environmental factors on the spawning day relative to the full moon for *Acropora spp.*

| | All data | | | Baa Atoll | | | North Male Atoll | | |
|---|---|---|---|---|---|---|---|---|---|
| | $\tau$ | z | p | $\tau$ | z | p | $\tau$ | z | p |
| **Total daily precipitation (mm)** | 0.276 | 12.162 | **<0.001** | 0.115 | 3.226 | **0.001** | 0.419 | 12.732 | **<0.001** |
| **Average daily wind speed (mph)** | 0.481 | 22.399 | **<0.001** | 0.246 | 7.079 | **<0.001** | 0.438 | 14.302 | **<0.001** |
| **Tide depth (m)** | 0.626 | 26.111 | **<0.001** | 0.724 | 18.882 | **<0.001** | 0.470 | 14.109 | **<0.001** |
| **Average daily sea surface temperature (°C)** | −0.295 | −13.746 | **<0.001** | 0.284 | 8.199 | **<0.001** | −0.354 | −11.554 | **<0.001** |

Notes.

Significant results are denoted in bold. Values less than 0.001 (highly significant) are denoted as <0.001.

### Reef type variations

Of the three species in Baa Atoll assessed for differences in spawn day relative to the full moon and spawning time after sunset between frame and wild colonies, no significant differences were found.

## DISCUSSION

The results from this study expand our limited knowledge on the exact spawning times of 22 *Acropora* spp. across two Maldivian atolls and identifies two peak spawning periods each year. The first season occurs between March and April, and the second

season occurs between October and December. Similar patterns have been observed in Western Australia (*Rosser & Gilmour, 2008*; *Rosser, 2013*; *Gilmour, Speed & Babcock, 2016*), Indonesia (*Permata et al., 2012*; *Wijayanti et al., 2019*), and Singapore (*Guest, 2005*), with peak spawning events inferred from recruitment studies in Sri Lanka during March–April (*Kumara, Cumaranatunga & Souter, 2007*). In addition, this study demonstrates spawning events occur in the Maldives over an extended period of eight months of the year (Table 1), which has similarly been seen in other equatorial reefs (*Gouezo et al., 2020*).

In North Male Atoll, more species were observed spawning in the first season (16 species) compared to the second seasons (six species), but in Baa Atoll, nine species were observed during each season (Table 1). The climate in the Maldives experiences a wet season, accompanied with the west to northwest winds associated with the *hulhangu* monsoon, from April to November and a dry season known as the *iruvai* monsoon associated with winds from the east-northeast from December to March (*Kench & Brander, 2006*). The transition period of the *hulhangu* monsoon takes place between March and April (*Aleem, 2013*), which appears to be associated with the largest number of multi-species spawning observed in North Male Atoll. In comparison, the *iruvai* transitional period from October to November (*Aleem, 2013*), signifying the end of the wet season, appears to show fewer species spawning in North Male Atoll. In contrast, nine species were observed spawning in each transitional period in Baa Atoll. Multi-specific spawning events have also been observed to take place during monsoonal transition periods in Indonesia (*Wijayanti et al., 2019*; *Indrayanti et al., 2019*). The bi-annual monsoon change strongly influences environmental parameters such as winds and currents in the North of the Maldives, in comparison to the South of the Maldives, which is less affected by monsoon changes and influenced by the equatorial currents (*Su, Wijeratne & Pattiaratchi, 2021*). Given this study documented differences between two atolls in relatively close latitudinal proximity, the contrasting influence of the bi-annual monsoon change from the north and south of the Maldives highlights that further inter-atoll differences could be observed in spawning patterns across the North-to-South atoll chain of the Maldives and may be linked to the different effects of the monsoon, as well as intra-atoll variations further influenced by local environmental factors. This emphasises the need to document and record coral spawning patterns across the Maldives and will help to expand our understanding of inter- and intra-atoll connectivity.

Some inter-atoll variability observed in spawning patterns can likely be explained by species composition at our study sites. Some species present at Landaa Giraavaru are not present at sampled sites in Furana Fushi, for example *A. rosaria*. Other species such as *A. retusa* are present at Furana Fushi but were not observed with gametes or spawning. Additionally, a recent study by *Davies et al. (2023)* found that corals exposed to light pollution are spawning between one and three days closer to the full moon compared to those on unlit reefs, for the majority of genera. Furana Fushi is situated near to both Male, the capital city of the Maldives, and Velana International Airport and thus exposed to greater artificial light pollution, which could explain the significantly earlier spawning days relative to full moon observed in North Male Atoll compared with Baa Atoll. Both the potential influence of light pollution and varying inter-atoll species composition further

emphasise the need for understanding the potential disparity in spawning patterns across the Maldivian atolls.

At both sites surveyed in this study, colonies of *Acropora* utilised multiple spawning events:

(i) 14 colonies spawned sections of their branches over consecutive days; (ii) several species utilised 'split spawning' and spawned in consecutive months within the same season; (iii) two colonies spawned over two lunar phases within the same month and; (iv) one colony was observed spawning in two consecutive spawning seasons. Corals utilise variations in spawning synchrony as a mechanism for reproductive isolation and to reduce interbreeding (*Gilmour, Speed & Babcock, 2016*). For one colony to spawn during two spawning seasons within one year remains rare, but has been documented in *A. tenuis*, *A. cytherea* and *A. florida* (*Gilmour, Speed & Babcock, 2016*). Utilising multiple spawning events, either over multiple days in the same month, multiple months in the same season, or two consecutive seasons, can have individual advantages; namely to: (i) increase the likelihood of successful fertilisation; or (ii) minimise the effects of a single catastrophic event on reproductive success (*Harrison et al., 1984*; *Babcock et al., 1986*; *Richmond & Hunter, 1990*), or (iii) to help realign reproduction events to favourable environmental conditions (*Hock et al., 2019*), particularly in instances of multi-specific spawning events or events in which the number of conspecific colonies spawning is high. Despite potential advantages, such mechanisms of 'split' spawning in *Acropora* corals occur periodically in the Maldives, which concurs with research from Australia (*Foster, Heyward & Gilmour, 2018*). This could be explained by the date of where the full moon falls in the lunar month each year. *Foster, Heyward & Gilmour (2018)* identified that split spawning took place regularly on Scott Reef when the full moon occurs in the first week of the typical spawning month or the last week of the previous month. However, in some documented cases, conspecific spawning in different seasons has the potential to impede gene flow and result in genetic divergence (*Rosser, 2015*; *Rosser et al., 2020*). Further research into the genetic structure of *Acropora* corals in the Maldives would be beneficial to distinguish the true extent of species-specific spawning across and within atolls over time.

Although both atolls experienced two peak spawning seasons in this study with similarities within those seasons in terms of species composition, differences observed between North Male and Baa Atolls in spawning seasonality, day, and time suggest the potential for further inter-atoll variation in *Acropora* coral spawning across the Maldives. Although we found local environmental factors are significantly correlated with spawning day relative to full moon events, our results show considerable inter-atoll differences in spawning seasonality within populations of the same species, with four species experiencing differences in spawning season between atolls. Therefore, we can hypothesise that across the Maldives there is variation between other atolls in *Acropora* spawning seasonality and synchrony.

The day of spawning relative to the full moon varied annually for five of the six species tested. Although this study did not attempt to establish a causal relationship between environmental conditions and coral spawning, our results demonstrate that the proximity of coral spawning to the full moon is significantly correlated with various environmental
conditions. While the strength of these correlations varied between atolls and between environmental variables, temporal and geographic variability can be expected based on local factors influencing the night of spawning. However, whether regional environmental conditions are influencing spawning seasonality across Maldivian atoll chains year-on-year remains to be explored.

Across multiple spawning events and both atolls, we recorded *A. tenuis* spawning, on average, 21.2 min after sunset, and earlier than other species during nights of multi-specific spawning (Fig. 3B). *A. tenuis* has also been observed spawning soon after sunset in Australia and Japan (*Harrison et al., 1984*; *Hayashibara et al., 1993*; *Fukami et al., 2003*). Shifting spawning time could be a mechanism to reduce the risk of hybridisation for broadcast spawners during mass, multi-specific spawning events (*Palumbi, 1994*; *Knowlton et al., 1997*; *Fukami et al., 2003*). Therefore, *A. tenuis* could be spawning earlier than other species to ensure fertilisation only between conspecifics. Additionally, the mechanism to shift spawning time could also explain significantly different inter-annual spawning times experienced by five species of *Acropora* in this study, due to the high number of species recorded spawning on the same night. During multi-specific spawning events, species can stagger their gamete release times through the evening to prevent hybridisation (*Fukami et al., 2003*; *Levitan et al., 2004*; *Van Woesik, 2010*). On average, *Acropora* species observed spawning in this study first released gamete bundles over 100 min after sunset, with the exception of *A. tenuis* and *A. valida* (Fig. 3B). *Fukami et al. (2003)* also observed *A. austera* spawning earlier than other mass-spawning species, which was not the case in our study where *A. austera* spawned, on average, 102 min after sunset. However, our average is based on 3 colonies, which could be anomalous.

This study provides the first insight into restored and wild colony spawning behaviour of *Acropora* in the Maldives. Neither the day or time of coral spawning in Baa Atoll showed a significant relationship between reef type (frame and wild) of *A. millepora*, *A. humilis* and *A. digitifera*. Due to a low sample size in wild colonies affected by previous bleaching events (*Pisapia, Burn & Pratchett, 2019*), this analysis could only be conducted on three species. Despite these sample size limitations, our study provides preliminary evidence that restoration activities may not impact the natural spawning cycle of *Acropora* at our study site and there is potential for cross-fertilisation between restored and wild colonies, thus aiding natural recovery by boosting reproductive output. A critical aim of coral restoration is to increase coral cover, diversity, and fecundity. Given the rate of decline of *Acropora* in the Maldives and subsequent popularity in restorative activities, the ability for frame colonies to reproduce sexually and fertilise with wild conspecifics can improve the resilience of reef ecosystems. *Zayasu & Suzuki (2019)* found greater genetic diversity in an artificial population of *A. yongei* compared with wild colonies, showcasing the ability of restoration efforts to improve resilience of coral reefs to future stressors. The preliminary findings of spawning patterns based on reef type in this study demonstrate the need to consider restored colonies when discussing sexual reproductive patterns of *Acropora* in the Maldives.

## CONCLUSION

This research details for the first-time exact spawning times of *Acropora* across two Maldivian atolls elucidating patterns of multi-specific spawning, asynchronous spawning and 'split spawning' across multiple lunar phases. It is clear the Maldives experiences two distinct spawning seasons throughout the year, but spawning events can occur over an extended period of eight months. Inter-atoll variations in spawning day are likely influenced by local environmental factors, however further research into coral reproductive patterns of multiple study sites within atolls and across the Maldives needs to be conducted to ascertain regional disparities and seasonal variations. The preliminary findings of spawning patterns based on reef type in this study demonstrate the need to consider restored colonies when discussing the sexual reproductive patterns of Maldivian *Acropora* in the future.

## ACKNOWLEDGEMENTS

This research took place at Four Seasons Landaa Giraavaru, Baa Atoll and The Sheraton Full Moon Resort and Spa, Furana Fushi, North Male Atoll. We would like to thank both properties for their continued support during the collection of this data and to the wider Reefscapers team for their assistance of data collection during coral spawning events throughout the two-year study.

### Funding

The authors received no funding for this work.

### Competing Interests

The authors declare the following potential competing interests: Margaux Monfared, Kate Sheridan, Simon Dixon and Matthew Gledhill were employed by Reefscapers during the conceptual design and analysis of this research. Thomas Le Berre is managing director of Reefscapers.

### Author Contributions

- Margaux A.A. Monfared conceived and designed the experiments, performed the experiments, analyzed the data, prepared figures and/or tables, authored or reviewed drafts of the article, and approved the final draft.
- Kate Sheridan conceived and designed the experiments, performed the experiments, analyzed the data, prepared figures and/or tables, authored or reviewed drafts of the article, and approved the final draft.
- Simon P. Dixon conceived and designed the experiments, performed the experiments, authored or reviewed drafts of the article, and approved the final draft.
- Matthew Gledhill performed the experiments, authored or reviewed drafts of the article, and approved the final draft.
- Thomas Le Berre conceived and designed the experiments, authored or reviewed drafts of the article, and approved the final draft.

## Data Availability

The raw data are available in the Supplementary Files.

## Supplemental Information

Supplemental information for this article can be found online at http://dx.doi.org/10.7717/peerj.16315#supplemental-information.

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
