# Peer review of "Coral spawning patterns of Acropora across two Maldivian reef ecosystems"

_PeerJ, doi:10.7717/peerj.16315_

## Round 0.1 · original submission · Major Revisions

I have completed the evaluation of the manuscript, and in agreement with the reviewers, who provided very helpful and specific comments, there are some issues (e.g., statistical methods, spawning observations) that should be addressed before its acceptance.

·

Basic reporting

no comment

Experimental design

no coment

Validity of the findings

no comment

Additional comments

The manuscript by Monfared et al. titled “Coral spawning patterns of Acropora across two Maldivian reef ecosystems” reports the results of 18 months of monitoring of wild and restored corals in two Atoll locations to investigate spawning timing and behaviour. The authors recorded the spawning of 1200 colonies across 22 species of Acropora corals, and related this to timing relative to the full moon, relative to sunset, and related to several environmental metrics (rainfall, wind, temperature and tide) and tracked this through time. In general the manuscript is well written with an excellent discussion, well contextualized within the literature, and the knowledge gap clearly identified. I also commend the authors for a clearly huge field campaign to collect all the data required.

I have three major concerns with the manuscript at present: (i) there is some confusion around the statistical methods and results, which don’t always seem to align with the tables and figures, thus making it difficult to assess the validity of the results and interpretation of the data; (ii) the methods used to make spawning observations and the scale at which these data are interpreted (sub-minute) requires revision; and (iii) the methods used for classifying reproductive stage based on pigmentation colour requires re-evaluation. I discuss each of these in turn, below, and suggest a revision to address these concerns prior to publication.

MAJOR COMMENTS:
- Statistical methods and results. In general, the GLM and GLMM approach used is valid, but the reporting of the results in the text, tables and figures doesn’t always seem to match, and thus it is difficult to assess the validity of the results and interpretation of the data based on what is reported. Some specific points below:
a. The results reported in tables 2 and 3 seem to include the intercept values for each species, in addition to the slopes, which are the informative values from these models. The significance of the intercept is not meaningful wrt the hypotheses being tested and the estimates don’t make ecological sense, so I would suggest removing these lines from the table.
b. Also, I’m confused why so many of the model estimates for spawn date relative to the full moon are negative up to -5 when you only monitored starting 2 days before each full moon? And what does a value of -24 in the year explanatory variable mean?
c. Most of your species estimates are also negative for spawning time after sunset (Table 3), but Fig 3B shows all estimates are well after sunset. This is confusing. Furthermore, in the methods it stats that observed entered the water prior to sunset and remained checking for up to 2 hours after sunset (line 184). How can many of the estimates in 3B be well over 120 min after sunset?
d. In Table 4, It is unclear whether species was considered as a random or fixed effect in these models (or neither)? Since timing varies by species, I think species needs to be accounted for in the models.

- Spawning observations. I have a few concerns about the spawning observation methods and the ability of observations to be made to the minute and be statistically compared. Many of the results are presented (and interpreted) to the decimal minute. However, I suggest re-contextualizing them within what would be ecologically relevant. For example, lines 262, 295. If Acropora tenuis spawned 1.9 min later per year, is this really ecologically significant? I would suggest that it is likely well within the error of the spawning observation timing itself, especially if each diver was monitoring multiple colonies and also free divers were monitoring the colonies (i.e. free divers couldn’t observe every minute continuously). Furthermore other environmental data not considered (such as cloud cover influencing the ‘experienced’ timing of sunset) could easily affect this 1-2 min variation. As another example, line 295 indicates wild colonies spawned 3.9 min earlier than frame colonies. Is this within the error of observations? Do divers do rounds of the frames and then the wild colonies and then circle back? This may be statistically significant but the ecologically significance isn’t necessary clear. Therefore, I would suggest moderating the way these results are presented and discussed. As another example, (line 270) corals in North Male Atoll may have spawned 26 min earlier overall, but as you subsequently point out, the data should be interpreted at the species level. Therefore, I suggest editing to “While corals in North Male Atoll spawned 26 minutes earlier than at Baa Atoll, this was largely due to variations in coral community composition and at the species level, within species spawning time was consistent, except for A. secale and A. plantaginea which spawned 7.6 min earlier and 7.0 min later in North Male, respectively.”

Secondly, Line 180-184. Several species of Acropora spawn more than 3 hours after sunset (i.e. 'the late spawners') (see Babcock et al. 1986 and the Coral Spawning Database). How can you be sure you didn't miss these late spawners if observations ceased at 2 hours past sunset?

- Reproductive stage classification. It has been well documented that oocyte pigmentation varies considerably amongst species and can also vary amongst genotypes within species (i.e. some individuals have less pigmentation at spawning than others). Colours also range widely, even within acroporids, and at maturity can be white, cream, tan, orange, red, pink, and yellow, amongst others (See Babcock et al. 1986, Table 1). Therefore, I do not think that that the colour scale used here (white – pale – pigmented) can be applied across species consistently to indicate level of reproductive maturity. However, I recognize the need to use an ordinal scale in such broad-scale monitoring situations. Thus, I can suggest the authors consider adding a size class to this grouping (i.e. white and < 200 um) that would help indicate stage of maturity (nothing that size is also species-specific). Secondly. It seems that the classification was used just to inform the subsequent in field monitoring for spawning, is this correct? If so, and as long as the maturation data are not used in any statistical analysis, I think you could re-frame this classification as an observational tool used to prompt spawning monitoring.

SPECIFIC COMMENTS
Please see annotated PDF for edits and specific comments throughout.
Methods:
- How many of the 1200 records were actually the same colony reported on multiple times? Please clarify how many total different (individual) colonies were used in the analysis. Also, since you have repeated measurements of the same colony over multiple months and years, I think the models may require a colony ID as a random effect in the models?
- Were your model diagnostics checked/validated for assumptions? Please include a description of model validation methods.
- What data distributions did you use in each of these models? I suggest you include a table of each analysis run so it is clear exactly how you analysed the data
- Environmental data. Why not calculate the tidal depth for the day and time of spawning? Would this not be more accurate than the depth of low tide closest to sunset on the day of spawning?
- Ensure all methods and results are written in past tense. For example line 237 should read ‘represented 22 species’ and Lines 290 and 293 should read “There was no significant difference…”. I have edited a few other instances. Please check throughout.
- Check significant digits in the values and stats presented in the methods and results and round appropriately (considering the resolution of the data collected).

Table 1.
- The 2nd and 3rd columns present the same information as the last two columns, just added together. They can be removed.
- Also adding a column of the range of spawning days after full moon next to the months of spawning would be a valuable addition to the summary table.

Figure 1.
- Please add a latitude/longitude marker for reference.
- Also, I would encourage the authors to include an inset map identifying where the area is within a more regional context, and also boxes around the two study regions on the main map, to show which areas of the atolls are shown in the larger maps.

Figure 3.
- It appears that these results the means and ranges of the original data. I would suggest you instead present the marginal means of the model estimates so you can include statistical significance on the figure and the error in the estimate is captured visually.
I note that there are no figures showing the results of the environmental predictors on spawning time (table 4). As this is an important point of discussion in the manuscript, I suggest plotting those model outputs for a figure.

I also note that when trying to run your R code, the Proximity_FM_neg and pos variables are not included in the excel file for analysis. From the code it appears that the glm model diagnostics were not run.

·

Basic reporting

In this manuscript the authors describe for the first time a comprehensive description of Acropora spawning times in two Maldivian Atolls. This provides an invaluable source of information to the coral reef community to those interested in coral reproduction, reef connectivity or utilising sexually reproduction as a source of material for reef restoration practises. Focusing on the Acropora genus is also highlight relevant given the precipitous declines seen in species of this genus across the region a result of climate driven bleaching events, and the urgent need to reverse this trend.
The manuscript has been well written with a clear, well-structured introduction that covers the rationale behind the study.
Literature references are appropriate within the context of the study. Suggests have been made in the comments below.
The figures, tables and RAW data included are all relevant to the manuscript
Suggested improvements are below:
Line 82 – Perhaps use more recent paper such as Lin et al 2021 https://www.pnas.org/doi/10.1073/pnas.2101985118 & Kaniewska et al 2015 https://elifesciences.org/articles/09991?
Line 125 – This assumes there is multiple genotypes of each species to ensure cross fertilisation. Perhaps clarify with the addition of “However, the successful out planting of vast
numbers of genets of the same species can lead to ‘spawning hubs’ that reproduce sexually….”
Line 304 – also Singapore has two spawning peaks linked to the vernal (larger peak) and autumnal (smaller peak) equinoxes https://www.researchgate.net/publication/256297506_Reproductive_patterns_of_Scleractinian_corals_on_Singapore's_reefs
Line 406 – suggest ending the sentence with - cross-fertilisation, thus aiding natural recovery by boosting reproductive output.
Line 423 – suggest changing ‘indicating observation’ for ‘elucidating patterns of’
Figure 2 legend Figure 2A-C add species shown. These all look like A millepora.
The description of Fig2 D & E are the wrong way round based on the pictures and referenced earlier (Line 189 & 193). Currently description of Fig2 D is moment of bundle release & Fig2 E is ‘bundling’ (see earlier comment on bundling vs setting).

Experimental design

The amount of field work to collect the coral spawning observations must have been considerable, with many hours of hours underwater, and due diligence taken to collect data in a manner that allows comparisons between the Atolls is evident. This has allowed for research question to be defining the reproductive timing of Acropora across two Atolls and investigating the environmental parameters that drive these patterns, to be answered.
The methods are clear and statistical analysis is robust, with appropriate models used.
A few minor suggestions to improve the methods section:
Line 155 … Change taxa to genus
Line 162 – As the study is comparing inter Atoll spawning times, I suggest adding the distance between the two locations (Furana fushi and Gulhifalhu) and clarify here that the donor and recipient site were within the same Atoll. I had to goggle this to ensure this was the case.
Line 169 – Were fragments removed from the centre of the colony, avoiding the infertile peripheral growth edge, rather than the base? See Randal et al 2021 - https://researchonline.jcu.edu.au/64858/
Line 170 – Do you use scissors for sampling? Wire cutters perhaps?
Line 184 – This is personal preference, but I prefer to see the description as setting, which is more commonly used, rather than bundling.

Validity of the findings

Given the lack of any previous extensive coral spawning observations, other than a handful in the Coral Spawning Data Base, within the Maldives this study is an important contribution to the field. In much the same way that Babcock et al 1986 is still used by many researchers working on coral reproduction on the GBR as a guideline to the spawning timings this manuscript now offers the same opportunity within the Maldives.
In addition, it provides statistical evidence that wild vs frame colonies have no significant difference in spawning time in relation to lunar phase and minutes post sunset (Line 290 – 295) and highlights the importance of restoration initiatives in supporting reef recovery and reestablishment of reproduction following allee effects from population declines.

Additional comments

Overall I think this is a very nice manuscript and one that will provide an important reference point for Acropora spawning timings within the region for many involved coral reproductive work. I’m sure it will also lead onto further work to understand spawning variations throughout the whole of the Maldives, North to South and more broadly cross the Indian Ocean. I thank the authors for their and work and asking me to review this manuscript.

---

## Round 0.2 · Minor Revisions

Dear Authors,
Together with the reviewer, I read the revisioned manuscript and the supporting letter and you have fulfilled most all the requirements made by the reviewers. Nevertheless, there are still some minor changes that need to be done before the paper will be ready for publication.

·

Basic reporting

no comment

Experimental design

no comment

Validity of the findings

no comment

Additional comments

I thank the authors for a thorough and considered response and revision, which I think has greatly improved the manuscript, and addressed the majority of my concerns. The spawning observations have been clarified, as have the methods used for classifying reproductive stage (along with the way in which those data were used). The only outstanding concern is regarding the statistical analyses. Once addressed (please see details below), this manuscript will make a very valuable contribution to the literature and, in particular, Table 1 and Figure 3 are going to be ‘go-to’ resources for years to come.

Methods:
- The assumptions of glms / glmms also include homogeneous variance across predictors and normally distributed errors. The modelling as it’s currently presented doesn’t test for all these assumptions. In doing so, using the DHARMa (residual diagnostics for hierarchical models) package and the data and code provided, it is clear that several of the models do not meet model assumption criteria (see below). A more thorough assessment of the models is needed to validate their appropriateness, and adjust as required. For example, the result of “simulateResiduals(PFM, plot=TRUE)” for the Proximity full moon model “PFM <- glm(newPFMN ~ Atoll, data = LGSH, family = poisson)” (figure attached) demonstrates issues with several diagnostics.
- Line 222: rather than a GLM, I think a better approach would be to use a GLMM where you can account for variance associated with year and/or species using random effects. For example, when you test the effect of atoll, include year and species as random effects to account for that variance. Similarly, when testing for the effect of year, include atoll and species as random effects. Or you could model year + atoll as additive fixed effects accounting for species as a random effect. This will enable you to identify responses more effectively.
- Line 225: I think you mean here that you used GLMs to compare spawning amongst reef types within each of 3 species from Baa Atoll. I don’t think this is a correlation analysis, as is written, rather it’s a comparison amongst treatment groups.
- Rather than ‘date’ I suggest using ‘day’ throughout the model descriptions and the results, just so it is clear that you’re referring to the day relative to full moon rather than the actual date. Instead of ‘spawning date proximity to full moon’ I suggest ‘spawning day relative to full moon’ in figure legends/captions. Also see lines 222, 237, 239, Fig3A caption, etc.
- The model estimates presented in the Table 2 appear to be on the log scale. I think these need to be back-transformed if you want to show them in units of day and minute, as indicated.
- Line 250: should the first correlation coefficient be negative here? And change ‘strong’ to ‘strongly’
- Lines 259-262: I am concerned about the removal of ‘outliers’ from the data simply because they caused overdispersion in the model. Were these formally detected using an outlier test? If so, this should be reported. If not, I’m not convinced their removal is justified. Because there were 35 colonies that all spawned 15-19 days after the full moon, these could represent a real and important, yet minor, spawning behaviour. There are several ways to combat overdispersion (i.e. by modelling the dispersion parameter or testing some other distributional families) that you may be able to try in this instance.

Minor comments
- Suggesting editing abstract to remove duplication of ‘local and regional scales’ within first 2 sentences
- Lines 57-60: It looks like the sessile adult stage/sedentary adult stage text is duplicated in this sentence. Please edit.
- Line 88: Suggest changing ‘subject to’ to ‘subjected to’
- Line 128: consider hyphenating ‘out-planting’
- Line 204: suggest adding ‘determine individual spawning times to the minute’
- Lines 299, 312: Please clarify whether ‘earlier’ means earlier in the night or earlier in the month.
- Line 315: change ‘has’ to ‘had’
- Line 325: this line is an incomplete sentence. Please edit.
- Line 327: edit text in the list so it reads ‘Western Australia, Indonesia and Singapore, with peak…’
- Line 328: add space after “(Guest, 2005), ”
- Line 337: delete the ‘;’
- Line 427: add ‘the’ between ‘with exception’
- Line 437: suggest ‘provides preliminary evidence that’ instead of ‘provides a preliminary trend indicating that’
- Line 437: remove the ‘;’
- Lines 440-442: this sentence seems redundant with the two sentences above. Consider removing or revising this section.
- Figure 3B – typo ‘after to sunset’

---

## Round 0.3 · accepted · Accept

Dear Authors,
I have reviewed the latest version of the manuscript along with the input from the reviewers. I am pleased to see the improvements made, and I am satisfied with the current state of the manuscript. Therefore, I am happy to inform you that the manuscript is now ready for publication in PeerJ.